**Cite this article:** Šigutová H, Šigut M, Kovalev A, Gorb SN. 2020 Wing wettability gradient in a damselfly *Lestes sponsa* (Odonata: Lestidae) reflects the submergence behaviour during underwater oviposition. *R. Soc. Open Sci.* **7**: 201258.

structural biology/ecology/behaviour

hydrophobicity, wettability, insect wings, nanostructures, Odonata, submerged oviposition

**Author for correspondence:**
Hana Šigutová
e-mail: hana.sigutova@osu.cz

# Wing wettability gradient in a damselfly *Lestes sponsa* (Odonata: Lestidae) reflects the submergence behaviour during underwater oviposition

Hana Šigutová[1], Martin Šigut[1], Alexander Kovalev[2] and Stanislav N. Gorb[2]

[1]Department of Biology and Ecology/ENC, Faculty of Science, University of Ostrava, Chittussiho 10, 71000 Ostrava, Czech Republic
[2]Department of Functional Morphology and Biomechanics, Zoological Institute, Kiel University, Am Botanischen Garten 1–9, 24118 Kiel, Germany

HŠ, 0000-0003-1134-248X; MŠ, 0000-0003-4876-9794; AK, 0000-0002-9441-5316; SNG, 0000-0001-9712-7953

The phenomenon of hydrophobicity of insect cuticles has received great attention from technical fields due to its wide applicability to industry or medicine. However, in an ecological/evolutionary context such studies remain scarce. We measured spatial differences in wing wettability in *Lestes sponsa* (Odonata: Lestidae), a damselfly species that can submerge during oviposition, and discussed the possible functional significance. Using dynamic contact angle (CA) measurements together with scanning electron microscopy (SEM), we investigated differences in wettability among distal, middle and proximal wing regions, and in surface nanostructures potentially responsible for observed differences. As we moved from distal towards more proximal parts, mean values of advancing and receding CAs gradually increased from 104° to 149°, and from 67° to 123°, respectively, indicating that wing tips were significantly less hydrophobic than more proximal parts. Moreover, values of CA hysteresis for the respective wing parts decreased from 38° to 26°, suggesting greater instability of the structure of the wing tips. Accordingly, compared with more proximal parts, SEM revealed higher damage of the wax nanostructures at the distal region. The observed wettability gradient is well explained by the submergence behaviour of *L. sponsa* during underwater oviposition. Our study thus

## 1. Introduction

Over millions of years of evolution, insect surface structures have been optimized to deal with different environments encountered during lifetime. Even terrestrial insects are often equipped with non-wetting surfaces to cope with the contact with water sources, condensed dew or rain [1–4]. Insect wings have been adapted for maximum flight performance while keeping lightness, stability and endurance [5]. Semi-aquatic insects (i.e. insects with a close relationship with water) usually have hydrophobic wings, as these large-surface areas can potentially be more affected by water adhesion. The same applies to long-winged insects which lack the ability to actively clean their own wings with their extremities [1,2,6]. Reduction of wettability is an adaptation which prevents wings from being overloaded by water droplets that can cause wing weight increase which may impair flying, and consequently, make the individual more difficult or unable to forage and more vulnerable to predators [7,8].

The phenomenon of wing wettability has been studied among insect orders [1,3,9–14]. A number of studies have reported the superhydrophobicity of the wings in certain taxa [1,2,7,8,12,14–18]. Superhydrophobicity occurs when a water droplet forms a contact angle (CA) greater than 150° with the surface; moreover, a surface should also display low degrees of CA hysteresis (CAH) and low sliding angle [19–21]. Such superhydrophobic wing membranes show a range of impressive properties—they do not only repel water droplets but also have self-cleaning and antimicrobial capabilities [3,7,17,18,22,23]. All hydrophobic surfaces (CA > 90°) can be enhanced to superhydrophobicity by the addition of a certain type of topography [20,24,25]. This may be the reason why superhydrophobicity of insect wing cuticle has been studied extensively from a morphological point of view [1,3,11,12,26–28]. However, often, superhydrophobicity is a result of a combination of surface topography, comprising features both on the micro- (greater than 1 µm) and nanoscale (less than 200 nm), and chemical heterogeneity at the micrometre scale [7,13,16,29–31].

Accordingly, superhydrophobicity of odonate (i.e. both damselfly and dragonfly) wings arises due to their membrane which is composed of nanostructured waxes creating stalk-like protuberances and filamentous rods. At a microscale, wing veins may additionally have spines enhancing superhydrophobic effect [1,7,9,13,16,29,32–36]. Odonates live on the edge between aquatic and terrestrial environment and superhydrophobicity enables them to thrive. In addition to all the above-mentioned advantages of being superhydrophobic, the main one is related to their reproductive behaviour. Females of some taxa oviposit endophytically (i.e. they insert their eggs into plant tissues), and can completely submerge when climbing down the submerged stems [37]. When underwater, superhydrophobic structures of their wings and bodies help them to create air pocket (plastron), holding air film steadily. Such 'physical gills' enable access to oxygen, and in addition to air retention, dissolved oxygen can diffuse to the air pocket from surrounding water, extending the individual's respiration capacity [21,33,38].

Odonate wings have been found to possess spatially heterogeneous wettability characteristics, with the surface micro- and nanostructure and topography playing a key role in determining these heterogeneities [29,34,35,39]. There is a number of studies focusing on the wettability of odonate wings in the context of application to industry or medicine, as they are valuable for the reconstruction of wing structures as a biomimetic template [1,7,16,29,30,34,35,39–41]. However, in ecological and/or evolutionary context, such studies are very rare, for example, the one dealing with sex- and age-related differences in hydrophobic structures of a damselfly *Calopteryx splendens* [36], or one examining submergence potential of *C. cornelia* [33]. Moreover, the latter type of studies is biased towards stream-dwelling species which oviposit in oxygen-saturated environment. Here, we aimed to examine spatial differences in the hydrophobicity of the wings of a pond-dwelling damselfly *Lestes sponsa* (Odonata: Lestidae), in which underwater oviposition may occur. Specifically, using dynamic CA measurements, we focused on the wettability of different wing regions, and on the differences between males and females. Additionally, by using scanning electron microscope, we investigated surface nanostructures potentially responsible for observed differences. Finally, the functional and ecological significance of spatially heterogeneous hydrophobic properties was discussed.

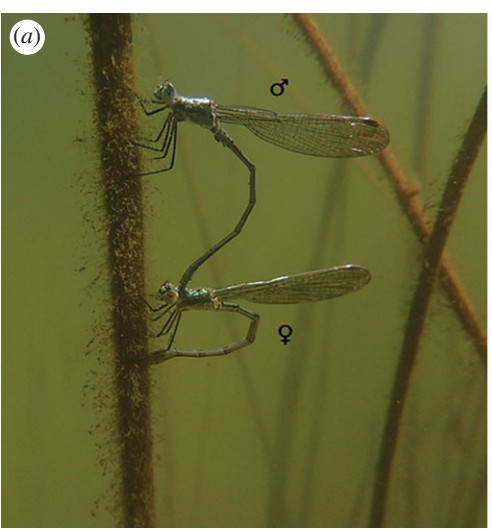 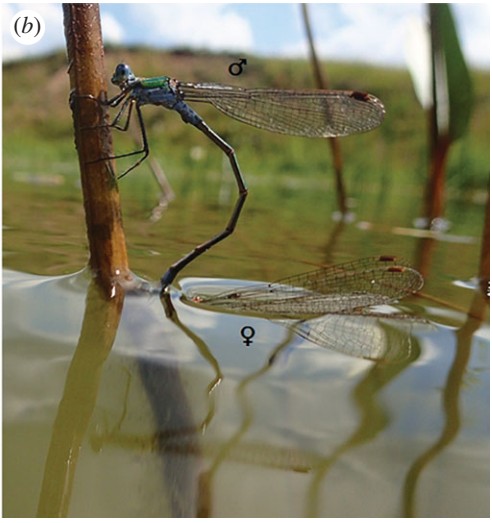

**Figure 1.** Tandem of *Lestes sponsa* during underwater oviposition while (*a*) submerged, (*b*) emerging.

## 2. Material and methods

### 2.1. Study organism

*Lestes sponsa* (Hansemann, 1823) is a widespread species ranging from Europe to northern Asia [42]. In this species, submerged oviposition is not a general strategy but it is applied at regional and local scales [43,44]. Females are able to lay eggs underwater into plant tissues in more than 50 cm depth and the duration of uninterrupted submerged oviposition is often longer than 20 min [43]. Males usually dive along with females as a tandem pair [37] (figure 1). Rarely, females lay eggs alone [45]. In August 2017, 66 individuals were collected from two localities where underwater oviposition has been previously recorded on a mass scale, both situated in the north-eastern part of the Czech Republic (32 from flooded limestone quarry—49.5886111° N, 18.1244444° E, 17 males, 15 females; 34 from fish breeding pond—49.6347878° N, 18.1012517° E, 19 males, 15 females). Even though in other damselfly species, the differences in hydrophobic properties between young and old individuals were negligible [36], we collected only actively ovipositing individuals to eliminate potential age-related differences. In the field, the damselfly individuals were collected in plastic containers and brought to the laboratory where they were immediately euthanized by freezing at −18°C. Specimens were then stored at the same temperature until measurements performed in October 2018. No specific permits were required to collect insect specimens, as *L. sponsa* is not protected in the Czech Republic. No specific permits were required for field sampling, as the sampled localities are not protected.

### 2.2. Contact angle measurements

CA between a surface and a water droplet refers to the hydrophobicity or hydrophilicity (wettability) of the surface [46]. However, in damselfly wings, significant spatial variations in CAs have been observed even at the microscale [29,34]. Therefore, to average CA over one direction (wing width) but preserve spatial resolution over other direction (wing length), CAs were calculated based on the force which had to be applied to immerse and withdraw the wing in/from the water (advancing and receding CA, respectively). Moreover, by using this method, we were able to obtain CAH which is another important characteristic of a solid–liquid interface occurring due to topographical and chemical heterogeneity [20,21,47,48].

To perform force measurements, the base of each individual's right hind wing (in individuals with damaged or incomplete right hind wing, the left one was used) was attached using screwing system to the tip of a force tranducer (FORT25, World Precision Instruments, Sarasota, FL, USA) mounted on a motorized micromanipulator (DC3001R, World Precision Instruments, Sarasota, FL, USA). Such an experimental design allowed us to measure forces acting on the wing during submersion and emersion. Only wings in good condition (no cracks or bending underwater) were used ($n = 66$), including 53 right wings (29 males, 24 females), and 13 left wings (7 males, 6 females). The wing was submerged (tip-first) into a transparent plastic container with 20 ml of distilled water at the speed of 0.2 mm s$^{-1}$. The wing

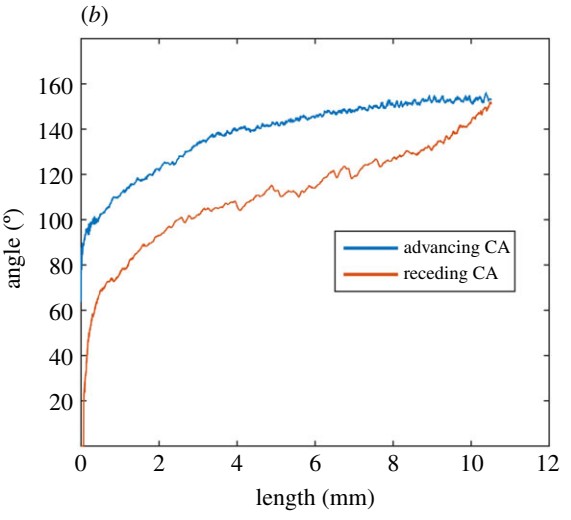

(a)

(b)

**Figure 2.** (a) Wing of *L. sponsa* mounted to the sensor during force measurements. Forces acting on the wing during submersion and emersion were measured and used for the calculation of advancing and receding contact angles ($\theta$), respectively. Average contact angle (CA) was determined in three different wing parts (distal, middle and proximal). (b) Example of CA calculations in Matlab. CAs were calculated along the whole length of the submerged part of wing.

was submerged to the depth of about 1 cm. When submerged, a photograph of the wing was taken using Canon EOS 20D digital camera (Canon Nanotechnologies, Inc., Austin, TX, USA) for later calculation of the submerged wing area. Subsequently, the wing was withdrawn from the water at the same speed. The force signals were amplified and measured using a MP100 data acquisition system and force–time data were recorded using AcqKnowledge (v. 3.7.3) software (both Biopac Systems, Inc., Goleta, CA, USA) with a sampling frequency of 200 Hz (figure 2a).

## 2.3. Scanning electron microscopy

To examine structural patterns of wax coverage, with regard to different levels of damage and presence of contaminants, scanning electron microscope (SEM) photographs were taken from three areas of the hind wings of selected individuals ($n = 6$). These areas corresponded with those used for CA measurements. Small pieces from each of the selected area were excised and mounted on aluminium holders using double-sided sticky conductive tape. The preparations were sputter-coated with gold–palladium (thickness 10 nm), and examined in a SEM Hitachi S-4800 (Hitachi, Ltd, Tokyo, Japan) at 3 kV. Photographs were taken at a magnification of 10 000, and of 50 000. We avoided photographing areas where there was damage caused by the handling of the wings by the experimentalist.

## 2.4. Data analyses

Calculations of CA were performed in Matlab v. 2015a [49] based on the following formula:

$$\theta = a\cos\left(\frac{F}{L\sigma}\right),$$

where $\theta$ is an effective CA of the wing with the water; $F$ is the force measured during wing submersion or emersion; $L$ is the perimeter of the wetted wing surface, where the force $F$ was measured; $\sigma$ is a distilled water–wing membrane surface tension. While pressing the wing into the water, the advancing CA was measured, while pulling the wing out of the water, the receding CA was measured. As the force was measured during the whole submersion and emersion process, CAs along the whole submerged wing length were calculated (figure 2b). CA hysteresis was determined as the difference between the advancing and receding CAs for corresponding wing positions [36].

Average CA in each wing was determined in three different regions: distal part ($0.5$–$0.9\delta$), middle part ($3.0$–$4.0\delta$) and proximal part of the wing ($7.0$–$7.5\delta$) (figure 2a), where the square root of pterostigma area was used as a reference length $\delta$ as it correlates with the size of the wing [50].

**Table 1.** Average values for advancing and receding contact angles and hysteresis calculated for individual wing parts (distal, middle, proximal) and sexes (male, female).

| wing part | female | ±s.e. | male | ±s.e. | total | ±s.e. |
|---|---|---|---|---|---|---|
| advancing CA (°) | | | | | | |
| distal | 104.5 | 3.9 | 104.1 | 4.2 | 104.3 | 4.1 |
| middle | 135.6 | 6.0 | 133.8 | 8.2 | 134.6 | 7.3 |
| proximal | 150.2 | 6.9 | 147.3 | 11.7 | 148.6 | 9.9 |
| total | 130.1 | 19.9 | 128.4 | 20.0 | 129.2 | 20.0 |
| receding CA (°) | | | | | | |
| distal | 68.0 | 16.0 | 65.4 | 26.7 | 66.6 | 22.5 |
| middle | 104.6 | 11.6 | 99.8 | 23.2 | 102.0 | 19.0 |
| proximal | 126.7 | 13.2 | 119.5 | 19.9 | 122.8 | 17.6 |
| total | 99.8 | 27.8 | 94.9 | 32.4 | 97.1 | 30.5 |
| hysteresis (°) | | | | | | |
| distal | 36.5 | 17.0 | 38.7 | 27.1 | 37.7 | 23.1 |
| middle | 31.0 | 9.8 | 34.0 | 16.9 | 32.6 | 14.2 |
| proximal | 23.5 | 10.6 | 27.8 | 11.6 | 25.9 | 11.4 |
| total | 30.4 | 13.9 | 33.5 | 20.1 | 32.1 | 17.7 |

All statistical analyses were performed in R v. 3.6.1 [51]. In total, we obtained data of average advancing and receding CAs from distal, middle and proximal parts of 66 wings (36 males, 30 females). To test the differences in advancing CA, receding CA and hysteresis among individual wing parts and sexes, we performed three separate generalized estimated equations (GEEs) using the function *geeglm* from the package 'geepack' [52,53] with gamma distribution of errors and log link function. In each model, particular wing was set as a group variable. In models for advancing and receding CA, we set up the AR(1) correlation structure accordingly to the order of submersion/emersion of the parts of individual wings. To perform *post hoc* multiple comparisons of advancing and receding CAs and hysteresis among individual wing parts, we used *lsmeans* function with Tukey method from the package 'lsmeans' [54].

# 3. Results

## 3.1. Contact angle measurements

Regarding all performed measurements, advancing CA ranged from 91.9° to 180.0° (mean = 129.2 ± 20), receding CA ranged from 2.6° to 153.9° (mean = 97.1 ± 30.5) and CAH from 5.0° to 103.3° (mean = 32.1 ± 17.7). Regarding individual wing parts, we found significant difference in advancing CA ($\chi^2$ = 1542.0, d.f. = 2, $p < 0.001$), receding CA ($\chi^2$ = 388, d.f. = 2, $p < 0.001$) and CAH ($\chi^2$ = 48.4, d.f. = 2, $p < 0.001$). Advancing and receding CAs were the lowest on the distal part of the wing and increased towards the proximal part of the wing (table 1, electronic supplementary material 1, figure 3a). Accordingly, CAH was the highest on the distal part of the wing and decreased towards the proximal part of the wing (table 1, electronic supplementary material 1, figure 3b). However, we found no significant differences between sexes either for advancing CA ($\chi^2$ = 1.57, d.f. = 1, $p = 0.210$), or for receding CA ($\chi^2$ = 1, d.f. = 1, $p = 0.350$), or CAH ($\chi^2$ = 1, d.f. = 1, $p = 0.330$).

## 3.2. Scanning electron microscopy

At the nanoscale, we observed wax rods of the different level of damage (electronic supplementary material 2). Intact wax was more frequently observed at the proximal and middle parts, whereas on the distal parts, the wax structures were more damaged and covered by dirt (figure 4).

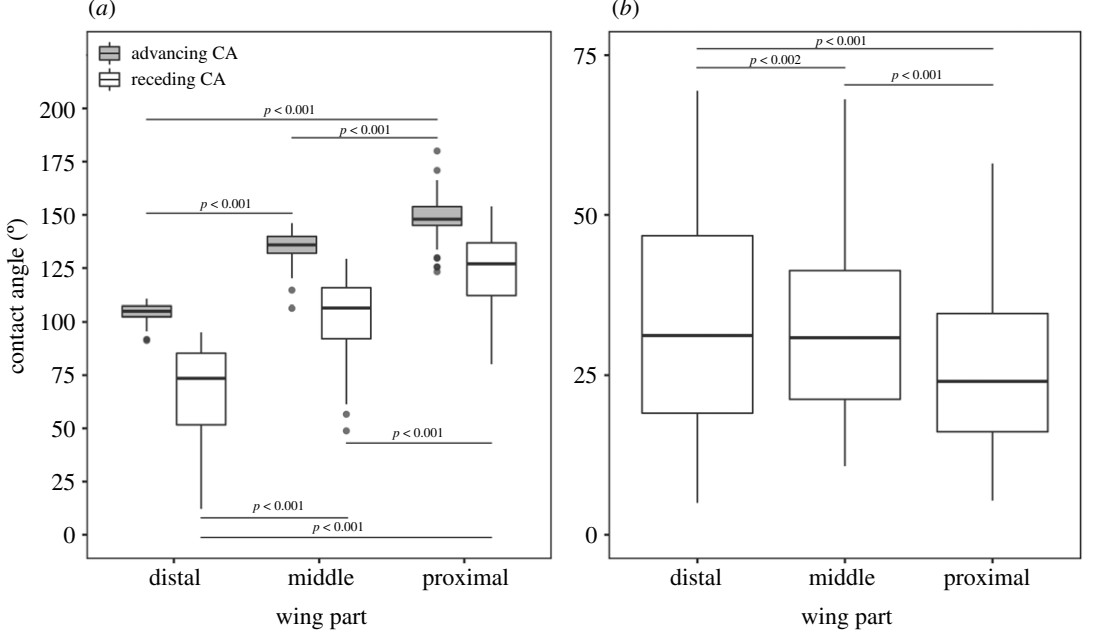

**Figure 3.** Contact angles measured among individual wing parts (distal, middle, proximal) of *Lestes sponsa*: (*a*) advancing and receding contact angles, (*b*) contact angle hysteresis.

## 4. Discussion

In accordance with our assumptions, we found significant spatial differences in the wettability of the *L. sponsa* wings. Contrary to the previously published studies on the wettability of different wing regions in both dragonflies and damselflies [35,39], we found the tip to be the least hydrophobic of all measured wing parts, and the hydrophobicity gradually increased towards the most proximal part. Unlike in some of the studies reporting CA values for the odonate wings [1,7,29,30,36], in *L. sponsa*, the wing surface as a whole was not superhydrophobic, as the mean values of advancing CA, receding CA and CAH for all measured regions were 129.3°, 97.1° and 32.1°, respectively, which was well below the superhydrophobicity threshold values (i.e. CA in excess of 150°). In general, all measured parts were rather hydrophobic (i.e. mean values of CAs ranging between 90° and 150°) [55,56], and in females, the advancing CA for the most proximal part even reached the superhydrophobicity threshold value. However, for a surface to be qualified as superhydrophobic, it must also display low CAH (<10°), and CAH for this part (=23.5°) still remained too high to indicate it as superhydrophobic as well as self-cleaning [21,57].

At this point, it is important to note that studies on the hydrophobicity of various odonate species [1,3,7,29,30,34–36,39] reported values of the static CAs obtained by using a sessile drop method [58,59]. In these studies, CA values ranged from 121° (*Gynacantha dravida*) [39] to 162° (*Orthetrum albistylum*) [1]. Nevertheless, measured values of static CAs depend on the volume of the droplet used in the experiment—larger droplet adds hydrostatic pressure onto the wing nanostructures which enables the droplet to penetrate further into the wing. Larger volumes of water thus tend to facilitate increased surface wetting [1,29]. Upon evaporation of the water droplet placed onto the wing surface, the CA decreases and stronger wetting occurs [29], which further complicates the interpretation of the results. The measured value of the static CA depends also on the wing part used for the measurement, as there is a large spatial variability in hydrophobicity of the wing surface, even at the microscale [29,34]. It is, therefore, difficult to compare the absolute CA values for different species measured among individual studies.

To eliminate the effect of evaporation and droplet volume, to average out local inhomogeneities of the surface, and due to the pronounced corrugated wing profile, in the present work, we measured dynamic CAs and hysteresis by using a force transducer. In the case of force measurements, CA values refer to the average over the whole immersed area of the wing [60], and thus take into account all potential scratches and irregularities. Advancing and receding CAs give the maximum and minimum values that the static CA can have on a flat surface [61,62]. Therefore, by theory, static CA values fall within the ranges of the

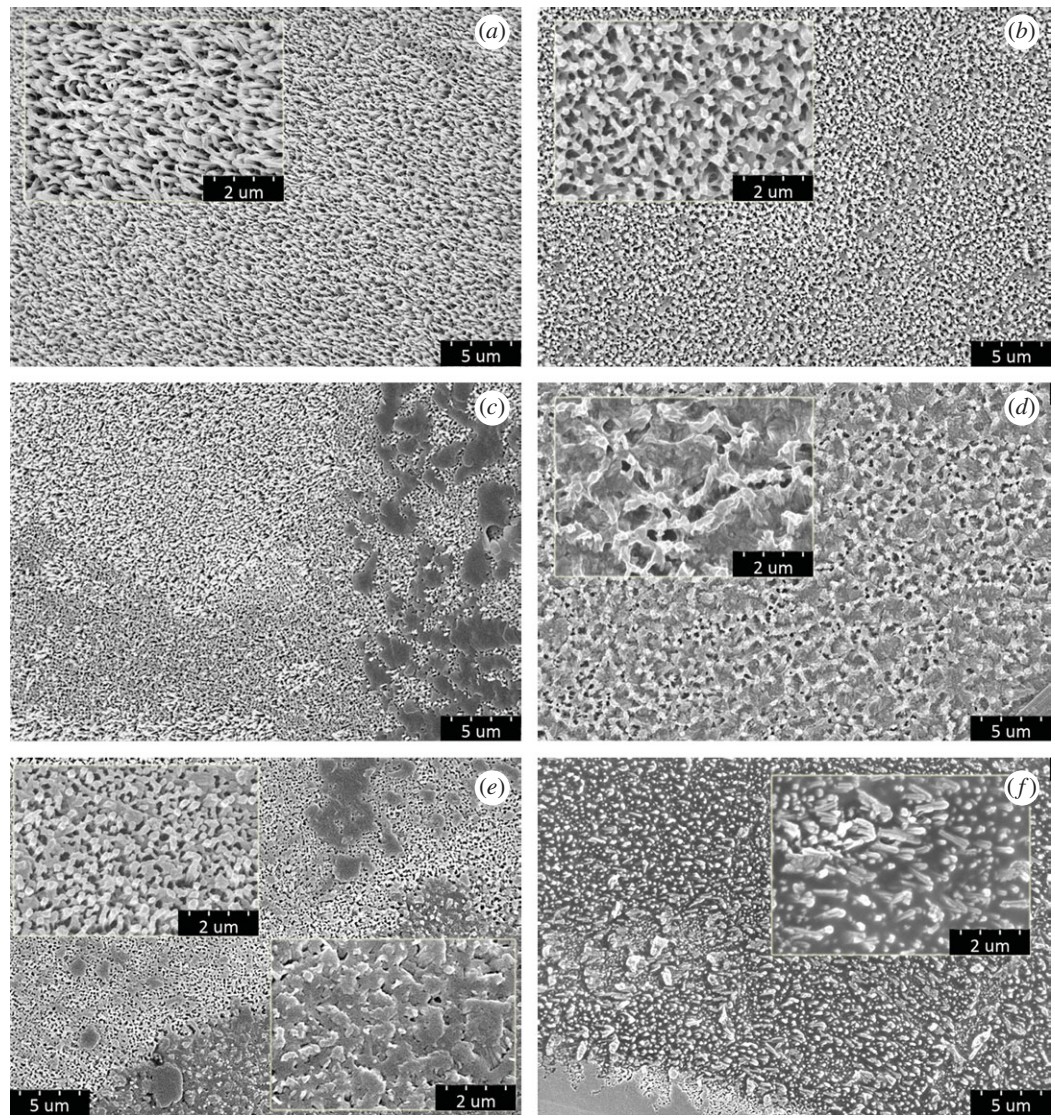

**Figure 4.** SEM photographs of the nanostructures of the wing membrane of *Lestes sponsa*. Wax rods of proximal and middle parts were either intact (*a*), or less frequently slightly joined together indicating slightly more damaged structure (*b*). Structure of the distal wing parts was either quite intact but dirty (*c*), or wax rods were strongly joined together, forming a damaged, network-like structure (*d*), or both damaged and dirty (*e,f*). Small sections (×50 000) represent details of large images (×10 000).

dynamic CAs. However, similarly to the static CAs, for dynamic CAs, results from different methods do not always agree and the method used should always be reported [21]. The possibilities of comparing the hydrophobicity of *L. sponsa* with other species studied are, therefore, limited.

There was a great range of values in all measured variables implying that there was a great variability in the hydrophobicity of individual specimens. According to Kuitunen *et al.* [36] there may be differences in hydrophobicity of the same individual between the dipping trials, as during the first dipping, dust particles present on the wings might be washed out, which increases hydrophobicity during the second dipping. Although we tried to obtain specimens of approximately the same age, they were sexually mature and they could have experienced different amounts of mating and oviposition events before being captured, which could have caused some damage to the wing structures [36]. Different number of submerged oviposition events could also have led to other differences, as cleaning of the wings due to the mechanism explained above could have occurred, or, conversely, the wings might have been contaminated with suspended solids which might have decreased their hydrophobicity, as water may adhere to these parts [3,9]. Moreover, when subjected to higher hydrostatic pressure, the natural surfaces may lose their hydrophobicity [63–65]. Our studied animals originated from the field where they encountered branches, weeds, etc. and their wing surface could have different amount of scratches and other damage which greatly affect wettability [34]. One way or another, the observed

differences in hydrophobicity of individual specimens point to the fact that such studies, if they are based on a small number of individuals, can lead to biased conclusions.

The significant hydrophobicity gradient found on the wings of the study species was quite straightforward. In Odonata, the observed differences in wettability of different wing regions are probably driven by differences in micro- and nanostructures, as evidenced both empirically and by theoretical modelling, whereas differences in chemical composition may play an insignificant role [39,62]. Superhydrophobicity of surfaces is generally explained by Cassie–Baxter (composite, heterogeneous) state of wetting, in which the air gets entrapped within the gaps between surface protrusions, reducing solid–liquid contact [66]. Simple hydrophobicity, on the other hand, corresponds to the Wenzel (wetted, homogeneous) state, in which these gaps are filled with water [67]. As pure Cassie–Baxter and Wenzel wetting situations rarely occur [68], more stable homogeneous transitional interface is common [65,69–71]. In our case, the observed variations in CAs among individual wing regions were probably due to the differences in the ability of water to penetrate between the cuticle features owing to their irregular order possibly caused by different level of damage and contamination by dirt. Due to the limited reach of the micromanipulator, it was not possible to submerge the wing entirely, including the most proximal part. Nevertheless, regarding the clear gradient from the wing tip to the proximal parts of the wing, one might expect that the hydrophobicity would further increase towards the base. Therefore, presumably, as we moved from the tip towards the base during the wing submersion, the transition occurred from the Wenzel state on more distal wing parts towards the prevalence of the Cassie–Baxter state on the most proximal parts.

This suggestion may be supported by the fact that CAH decreased from the wing tip towards the most proximal parts of the wing. Accordingly, hysteresis increases for a Wenzel state of wetting and decreases for a surface with Cassie–Baxter wetting [62,72,73]. By theory, CAH arises from the chemical and topographical heterogeneity of the surface, or rearrangement or alteration of the surface by the solvent [61,62]. Therefore, greater hysteresis may indicate greater surface instability. Moreover, on surfaces with complex topography, CA is usually higher and CAH is lower than on smooth surfaces [21]. Although quantifying the differences in surface structures of *L. sponsa* wings was beyond the scope of our study, based on the SEM pictures (figure 4, electronic supplementary material 2), we may conclude that distal wing parts possibly have less stable and less uniform surface structure than more proximal parts. Moreover, the level of corrugation of the wing varied among wing parts: at the basis, it was stronger; that is why it generated stronger and more stable air pockets than at the tip.

With the biological evolution, structure and materials of insect wings have been optimized to serve multiple purposes; the observed wettability gradient thus can potentially have a number of functions. Firstly, it may be helpful during submerged oviposition. This special type of oviposition strategy is associated with high risks and energy costs but can bring many benefits. Some hypotheses address avoidance of sexual harassment or male guarding behaviour, others protection of eggs from high temperatures, drying out or predation/parasitism, or exploiting additional oviposition sites in underwater substrates [74–77]. The way the female (and in the case of contact guarding also the male) gets underwater varies. Whereas some damselfly species proceed head-first (e.g. *C. splendens* or *Enallagma hageni*) [74,78], in *L. sponsa*, female begins the egg-laying process above the water level, and continues by moving backwards underwater, pulling the male with her [43,78,79]. When going underwater, tips are usually the first part of the wings that dips into water. As the tandem pair climbs down the stem, water level moves along the lower edge of the wings until the wings submerge completely, with proximal parts going first (JB Helebrandová 2019, personal communication). As the average CA of the wing tips was close to 90°, the energy needed to move them into and out of the water is minimal due to the small drag [70]. In the case of more hydrophobic tips, it would be difficult to submerge, as they would bend and slide along the water surface. Submerging with more hydrophobic proximal parts first is less complicated, as bending and folding is prevented by their connection with thorax. When climbing up, the wing tips are usually the first part of the damselfly body to emerge (figure 1*b*). Often, a short-term emergence before a subsequent submersion occurs, during which the male pulls his wing tips out of water to replenish the oxygen within the joint air bubble [79]. After replenishment, due to the more hydrophilic tips, it might be easier to get back underwater to continue with the oviposition, without making an unnecessary effort. Hydrophobicity gradient on the wings may be advantageous also during water–air transition during final emergence. The whole process of passing through the water level is usually very short [79]. Although it might require higher effort to push the wing tips through the water level at the beginning due to the surface tension, then, as an emerging individual proceeds, the wings come out of water more and more easily, without sticking to it, which may afterwards increase chances for quick flight start and

survival. In our study, the non-significant differences in hydrophobicity of males and females are not surprising, as both sexes submerge in the same way during oviposition, and both must equally resist rain and water splashes. The hydrophobicity gradient might also be useful to maintain certain body posture underwater, as is the case of dytiscid beetles [80].

Other possible function of hydrophobicity gradient could be relief from the load of water droplets in a directed manner, not dissimilar to directed wetting, a phenomenon known from beetles or butterflies [81,82]. In a damselfly *Ischnura heterosticta*, bouncing, airborne droplets on the wings tend to be guided towards more hydrophilic locations [29]. In *L. sponsa*, the observed hydrophobicity gradient could act as self-cleaning mechanism whereby a wing surface remains pristine due to the moving water droplets which, as they roll off the surface, collect any dirt and contaminants [20,41,83–85], at least at more proximal areas. As such droplets move towards less hydrophobic parts, their load of contaminating particles may tend to cling to the wing surface. Besides, on more hydrophilic surface water droplets remain for a long time until they evaporate. During the evaporation process, strong surface tension forces are acting on the surface structures and may potentially cause them damage which may further reduce their hydrophobicity. Consequently, the structure of the wing tips should be more damaged and more contaminated. This pattern was again evidenced by the SEM. Regarding the origin of our study specimens, studies with pristine wings of juvenile individuals with undamaged and uncontaminated wing structure could help to get better insight into the role of contaminants in determining hydrophobicity gradient.

It is possible that in zygopteran species applying different oviposition strategy as well as in species from the suborder Anisoptera which are capable fliers and with some exceptions do not submerge during oviposition [37,86], the opposite gradient could be found, as suggested by two recent studies [39,62]. In such case, any contaminants would tend to cling to the basal parts of the wings. Such situation could be advantageous in terms of better flight capabilities, as increased load at the wing tip may negatively influence flight mechanics, especially in long-winged insects [87,88]. Therefore, it is possible that in species that commonly apply submerged oviposition by moving backwards underwater, advantages of the gradient from less hydrophobic distal wing parts towards more hydrophobic proximal parts may outweigh impaired flight capabilities, especially in poor flyers such as *L. sponsa*. Our study proposed the existence of species-dependent hydrophobicity gradient on odonate wings, which might be caused by different ovipositional strategies. More studies with other species representing different modes of getting underwater and employing different type of male guarding behaviour may help to get further insight into processes shaping the evolution of insect wing surfaces.

Data accessibility. Original data supporting our article is provided as electronic supplementary material.

Authors' contributions. H.Š. carried out the experimental work, participated in the design of the study and drafted the manuscript; M.Š. carried out the experimental work and statistical analyses, participated in the design of the study and helped draft the manuscript; A.K. participated in data processing, participated in design of the study and critically revised the manuscript; S.N.G. coordinated the study, participated in the design of the study and critically revised the manuscript. All authors gave final approval for publication and agree to be held accountable for the work performed therein.

Competing interests. We declare we have no competing interests

Funding. This work was supported by the Erasmus+traineeship programme (KA1).

Acknowledgements. We thank our colleague Jana Branwen Helebrandová for sharing her detailed knowledge of ovipositional behaviour of *Lestes sponsa*, for providing photographs of ovipositing tandem pairs, and for her inspiring thoughts. We are also thankful to Esther Appel for her valuable advice regarding the practical aspects of conducting the experiment, and to Halvor T. Tramsen for designing and modelling of the wing holder for the attachment system.

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
