## [Reviewer comments · Royal Society Open Science]

Review History

RSOS-201258.R0 (Original submission)

Review form: Reviewer 1

Is the manuscript scientifically sound in its present form?

Yes

Are the interpretations and conclusions justified by the results?

Yes

Is the language acceptable?

Yes

Do you have any ethical concerns with this paper?

No

Have you any concerns about statistical analyses in this paper?

No

Recommendation?

Accept as is

Comments to the Author(s)

I appreciate the detailed responses to both reviewers' comments. They addressed my previous concerns and suggestions.

Decision letter (RSOS-201258.R0)

Dear Dr Šigutová:

It is a pleasure to accept your manuscript entitled "Wing wettability gradient in a damselfly *Lestes sponsa* (Odonata: Lestidae) reflects the submergence behaviour during underwater oviposition" in its current form for publication in Royal Society Open Science. The comments of the reviewer(s) who reviewed your manuscript are included at the foot of this letter.

on behalf of Professor Brooke Flammang (Associate Editor) and Professor Kevin Padian (Subject Editor).

Associate Editor Professor Brooke Flammang Comments to Author:

Reviewer(s)' Comments to Author:

Reviewer: 1

Comments to the Author(s)

I appreciate the detailed responses to both reviewers' comments. They addressed my previous concerns and suggestions.
